# Research Progress of the Biosynthesis of Natural Bio-Antibacterial Agent Pulcherriminic Acid in *Bacillus*

**DOI:** 10.3390/molecules25235611

**Published:** 2020-11-28

**Authors:** Siqi Yuan, Xihao Yong, Ting Zhao, Yuan Li, Jun Liu

**Affiliations:** 1Sichuan University of Science & Engineering, Xueyuan Street 180#, Huixing Rd., Zigong 643000, China; yuansiqi@suse.edu.cn (S.Y.); yongxihao@suse.edu.cn (X.Y.); zhaoting@suse.edu.cn (T.Z.); 2Luzhou Laojiao Group Co. Ltd., Airentang Square, Jiangyang District, Luzhou 646000, China; 3Wuliangye Group Co. Ltd., No. 150 Minjiang West Road, Yibin 644000, China

**Keywords:** pulcherriminic acid, *Bacillus*, bacteriostasis, key enzymes, synthetic pathway, catalytic mechanism, regulatory mechanism, metabolic engineering

## Abstract

Pulcherriminic acid is a cyclic dipeptide found mainly in *Bacillus* and yeast. Due to the ability of pulcherriminic acid to chelate Fe^3+^ to produce reddish brown pulcherrimin, microorganisms capable of synthesizing pulcherriminic acid compete with other microorganisms for environmental iron ions to achieve bacteriostatic effects. Therefore, studying the biosynthetic pathway and their enzymatic catalysis, gene regulation in the process of synthesis of pulcherriminic acid in *Bacillus* can facilitate the industrial production, and promote the wide application in food, agriculture and medicine industries. After initially discussing, this review summarizes current research on the synthesis of pulcherriminic acid by *Bacillus*, which includes the crystallization of key enzymes, molecular catalytic mechanisms, regulation of synthetic pathways, and methods to improve efficiency in synthesizing pulcherriminic acid and its precursors. Finally, possible applications of pulcherriminic acid in the fermented food, such as Chinese Baijiu, applying combinatorial biosynthesis will be summarized.

## 1. Introduction

Pulcherriminic acid is derived from cyclo (*L*-Leu-*L*-Leu) (cLL), a cyclic dipeptide synthesized by *Bacillus* and yeast [1,2,3]. The hydroxamic acid group with pulcherriminic acid structure chelates four Fe^3+^ to form a reddish brown pulcherrimin, thereby the microbes producing pulcherriminic acid can still accumulate iron ions in a severely iron-deficient environment. Since the concentration of iron in the natural environment is relatively low, microorganisms capable for synthesis of pulcherriminic acid compete with other microbes around them for iron through the high affinity of pulcherriminic acid to ferric iron in the environment, and occupy a competitive advantage, making other microorganisms unable to obtain the necessary inorganic salts for growth, especially the iron, which indirectly proves the potential antibacterial activity of pulcherriminic acid [4,5,6,7]. Therefore, Pulcherriminic acid is a very good and safe bioactive antibacterial material in industry.

Pulcherriminic acid is soluble in water, and its molecular formula and molecular weight are C_12_H_20_N_204_ (Figure 1) and 256.2980, respectively, which has maximum UV absorption peaks at wavelengths of 243 nm, 282 nm, and 410 nm [8]. The molecular formula of pulcherrimin is C_12_H_18_N_204_Fe_2/3_ (Figure 1), and it is reddish brown. Pulcherrimin is almost insoluble in water and organic solvents such as ethanol, but it can be dissolved in alkaline methanol or NaOH solution [8]. The red precipitates gradually appear as the pH of the solution decreases [9]. Pulcherriminic acid was first discovered in the fermentation process of yeast, and it has been found subsequently in *Bacillus subtilis*, *B. cereus*, *B. licheniformis,* and *Michia* [9,10,11].

In recent years, the synthetic pathways of pulcherriminic acid have been studied extensively in *Bacillus*, which eliminated the obstacles to increase its production. In 2006, Tang et al. [12] firstly described the synthetic pathway of pulcherriminic acid in *B. subtilis***,** and Gondry et al. [13] showed that YvmC protein in *Bacillus* was a member of the cyclodipeptide synthases (CDPSs). Bonnefond et al. [14] analyzed the entire synthetic pathway of pulcherriminic acid in *Bacillus* and proposed the pathway (Figure 1). Leucine (Leu) is a precursor of pulcherriminic acid (Figure 2I), which is synthesized under the catalysis of a series of enzymes with pyruvate as the substrate. These enzymes included acetohydroxyacid synthase (IlvBH), dihydroxy acid dehydratase and keto acid reductase (IlvC and IlvD), 2-isopropylmalate synthase, 3-isopropylmalate dehydrogenase and dehydratase (LeuABCD, Figure 2II) [15]. Leuyl-tRNA synthetase (LeuS) catalyzes the conversion of Leu into leucine-tRNA (Leu-tRNA^Leu^) [16]. Subsequently, two Leu-tRNA^Leu^ molecules underwent dehydration condensation to form cLL under the catalysis of CDPSs YvmC. Under the catalysis of the cytochrome P450 oxidase (CypX) that was expressed by the cytochrome P450 synthesis gene *cypX* downstream of the CDPSs gene *yvmC*, the formed cLL was finally oxidized to form pulcherriminic acid (Figure 2I) [14]. Finally, the major facilitator superfamily (MFS) transporter (YvmA) transported pulcherriminic acid out of the cell [17,18], and the pulcherriminic acid excreted from the cell was then combined with Fe^3+^ to form water insoluble pulcherrimin (Figure 2II) [17].

The mechanism for synthesis of cyclic dipeptides has been studied in depth. In *Bacillus*, YvmC and CypX are involved mainly in the synthesis of pulcherriminic acid, but reports on the transcriptional regulation of *yvmC-cypX* operon and the secretion of products are still needed further exploration. Due to the results of bacteriostatic studies, it showed that pulcherriminic acid had obviously bacteriostatic activity and could have great potential for applying in the fermented food, insecticides, fruit and vegetable fungicides, new pesticides, and other antibacterial drugs. Surprisingly, pulcherriminic acid and its intermediates or analogs have been determined in Chinese Baijiu (unpublished data). It is a distilled liquor produced by solid state fermentation of natural microorganisms (such as *Bacillus* and yeasts), thus, the traceability of the functional factors, such as pulcherriminic acid, will become another hot topic for the study of composition of Chinese Baijiu. Overall, clarification of the catalytic and regulatory mechanisms of the key enzymes for the synthesis of pulcherriminic acid could promote the application of pulcherriminic acid in agriculture, food protection and medicine.

The aim of this review is to summary the synthesis pathway and regulatory mechanism of pulcherriminic acid in *Bacillus*, to illuminate the current research status of the key enzymes in the synthesis of pulcherriminic acid improving the fermentation production, and to pinpoint several specific directions for future research. We hope that this review will provide some guidance for future research on the molecular and regulatory mechanisms for synthesis of pulcherriminic acid.

## 2. Cyclic Dileucine Synthase is Used to Synthesize the Precursor cLL of Pulcherriminic Acid

### 2.1. Cyclic Dipeptides

Cyclic dipeptides (also known as cyclodipeptides, 2,5-diketopiperazines or 2,5-dioxopiperazine, CDPs) are the smallest cyclic peptides formed by the condensation of two amino acid molecules through peptide bonds [20,21,22]. They belong to a class of a derivative family known as diketopiperazines (DKPs). DKPs belong to a large class of compounds in nature. Due to their stable six-membered ring structure, DKPs can be used as molecular fragments for other biosynthetic applications, such as molecular drug design [22,23], for antibacterial [24,25], antifungal [26,27], antiviral [28], antitumor [29,30], immunosuppressive [31], neuroprotection [32], antimalarial [33], anti-prion [34], and antihyperglycemic [35] applications, and for scavenging free radicals [36] and other significant physiological/pharmacological activity. With continuous research on DKPs, the biosynthetic pathways of several DKPs have been elucidated. These biosynthetic pathways are generally divided into non-enzymatic and enzymatic pathways. In the non-enzymatic pathway, the cyclization of the peptide chain is spontaneous without the participation of enzymes, such as cyclo(*L*-His-L-Pro), which is ubiquitous in the central nervous system of higher mammals [37]. Enzyme-catalyzed pathways can be divided into non-ribosomal peptide synthetases (NRPSs) and cyclic dipeptide synthases (CDPSs) [22,38,39]. Based on the current studies, most DKPs are synthesized by NRPSs, which include an adenylation domain, a peptidyl carrier protein domain, and a condensation domain [40].

In recent years, various cyclic dipeptides produced by microorganisms have been reported [41,42]. For example, cyclo (Phe-Pro), cyclo (Tyr-Phe), and cyclo (Leu-Iyr) molecules were used as signal sensors; cyclo (*D*-Pro-*L*-Phe) exhibited antibacterial activity, and cyclo (*L*-Phe-*L*-Pro) and cyclo (*L*-Phe-trans-4-OH-Pro) exhibited antifungal activity. cLL is a kind of cyclic dipeptide. It is a small molecule organic cyclization with high stability, protease resistance, and structural rigidity formed by the condensation of two molecules of leucine [22,37,43,44].

### 2.2. Cyclic Dipeptide Synthases (CDPSs)

In 2002, Lautru et al. [45] discovered an AlbC protein expressed by the *albC* gene in the genome of *Streptomyces noursei*, which catalyzes the production of cyclic dipeptides by a non-NRP pathway. Subsequently, Gondry et al. [13] firstly proposed the CDPS pathway [13,45]. In vitro AlbC enzyme activity studies showed that AlbC used aminoacyl tRNAs (aa-tRNAs) as a substrate to synthesize peptide bonds of DKPs, and in silico analysis showed that a variety of bacterial phyla contained AlbC-like proteins. Gene alignment showed that these AlbC-like proteins contained 13 conserved amino acid residues and were divided into two small regions: HX [LVI] [LVI] G [LVI] S and Y [LVI] XXEXP [46]. With the continuous research on CDPSs, we have a deeper understanding of their classification and structure [47].

CDPSs are usually small molecular proteins that consist of 200–300 amino acid residues. Most CDPSs have low sequence homology and <30% amino acid identity [37]. As of 2017, about 800 CDP-encoding genes had been identified through a search of sequence homology, and their number is steadily increasing [41]. They are distributed in all domains, but mainly concentrated in three bacterial phylum (*Actinobacteria*, *Firmicutes,* and *Proteobacteria*) [41]. Among them, about 400 CDPSs originated from *Actinomycetes* [48,49]. CDPSs are divided into NYH and XYP sub-families based on their Rossmann folds [50]. All CDPSs that have been discovered so far where catalytic function and/or structure have been elucidated belong to the NYH family [50,51]. 

In 2018, Bourgeois et al. [51] analyzed the CDPSs of the XYP sub-family and compared the structures of the representative enzymes (AlbC and Rgry) in the two subfamilies (NYH and XYP family). The main difference between the two structures was the first half of the Rossmann fold, (Figure 3I–III) [51]. The spatial positions of key catalytic residues (S37, Y202, E182) were the same in the two subfamilies, which indicated that the catalytic mechanism of the two families was probably the same [51]. There was a deep surface-accessible pocket (P1) in the CDPS structure, which was bounded by catalytic residues, and the conservative catalytic residues in its pockets also exhibited good overlap [38,41]. The structure of CDPSs was similar to the type I aminoacyl-tRNA synthetase catalytic domain (class-I aa-tRNA synthases, aa-tRSs), and it was especially similar to class-Ic TyrRSs and TrpRSs [46,52]. The structural differences between CDPSs and aa-tRSs were: (i) CDPSs did not have tRNA-binding domains in type I aa-tRSs, and they relied on surface amino acid residues to bind tRNA, (ii) TyrRSs and TrpRSs generally form homodimers, and the two active sites were in polymer interface, and (iii) the monomers of CDPSs were active [41,50].

The structure of the amino acid binding part of CDPSs was similar to that of aa-tRSs, so they had similar mechanisms for catalyzing the formation of peptide bonds. In particular, the position of the surface accessible P1 pockets in CDPSs was similar to the aminoacyl-binding pockets in aaRS (class-Ic TyrRSs and TrpRSs) [14,16,46]. A possible catalytic mechanism for CDPSs involved using two molecules of aa-tRNA as a substrate, and a cyclic dipeptide was synthesized by a continuous ping-pong reaction [53]. Taking AlbC as an example, AlbC hijacked Phe-tRNA^Phe^ and Phe-tRNA^Phe^ (or the second Leu-tRNA^Leu^) in turn [13] and used them as a substrate for the ping-pong mechanism (Figure 4). This involved the formation of two consecutive acylase intermediates [53]. The first Phe-tRNA^Phe^ bound to an enzyme so that its aminoacyl moiety was contained in the P1 pocket [16]. This portion was then transferred to a conserved serine residue (Ser37) to form the aminoacyl-AlbC intermediate. The second Phe-tRNA^Phe^ bound to the enzyme so that its aminoacyl moiety was located in a wide pocket P2, close to P1 [53]. Then, phenylalanyl-AlbC is reacted with a second Phe-tRNA^Phe^ to form a dipeptidyl-AlbC intermediate, and the final cyclic dipeptide is obtained by intramolecular cyclization. The formation of the first intermediate was demonstrated by trypsinization and peptide fingerprinting (PMF) on AlbC incubated with Phe-tRNA^Phe^; the second intermediate was captured by modifying a conserved tyrosine (Tyr202) involved in the cyclization process, and this intermediate was detected by PMF analysis using a Tyr202Phe mutant [53]. The catalytic residues identified in the P1 pocket were: Tyr178, Glu182, Asn40, and His203 [53]. 

Recent studies have shown that the final step of cyclization of cyclic dipeptide depended on the conserved tyrosine residues of CDPSs. As a proton relay, tyrosine residues catalyzed the cyclization reaction through the proton transfer of their hydroxyl groups [51,54]. In 2018, Schmitt et al. [54] studied the cyclization reaction mediated by the cyclic dipeptide synthase AlbC using the quantum mechanics (QM)/molecular mechanics (MM) method and free energy simulation. Experimental results revealed that Tyr202 existed in a neutral protonated form and was unlikely to participate in the catalytic process as a general base; because tyrosine has the unique structure of the benzene ring, the hydroxyl group of tyrosine was more suitable for the proton relay than the hydroxyl group of other amino acids, and the proton was unlikely to transfer directly from the amino group to Ser37 of AlbC [54]. In addition, that the residues Glu182, Tyr178, Asn40, and His203 maintained the correct dipeptide conformation required for the cyclization reaction [54]. More importantly, the functioning of this mechanism depended on the conserved amino groups in the entire CDPS family, and it may be used commonly in CDPSs [54].

The recognition of substrates by CDPSs depended on the type of amino acid and the molecular structure of the corresponding aa-tRNA, which resulted in low substrate specificity of CDPSs [13]. Therefore, several cyclic dipeptide by-products synthesized during the formation of cyclic dipeptides that were catalyzed by CDPSs significantly reduced the synthesis efficiency of the target cyclic amino acids and related metabolites [50]. At present, the structure and catalytic mechanism of some CDPSs have been determined through the analysis of crystal structure, but the kinetic parameters of CDPSs are still unstudied. Through kinetic analysis and structural analysis, it will lay a solid theoretical foundation for the improvement of substrate specificity of CDPSs.

### 2.3. Cyclic Dileucine Synthase (YvmC)

The AlbC-like protein YvmC, which is synthesized by *B. subtilis* and *B. licheniformis*, can use aa-tRNA as a substrate to synthesize cLL-based DKPs [13]. In 2011, Luc Bonnefond et al. [14] obtained the crystal structure of YvmC from *B. licheniformis* (*B. licheniformis* CDPSs, YvmC-Blic). X-ray diffraction experiments showed that the YvmC protein of *B. licheniformis* also had an aaRSs-like catalytic domain (especially class-Ic TyrRSs and TrpRSs) [46,55], which consisted of α / β folds with six stranded spiral units. However, unlike aaRS, the ATP binding motif present in aaRS did not exist in YvmC-Blic, and YvmC-Blic active unit was a monomer instead of a dimer. In addition, YvmC-Blic lacked the last helix and the helical inserts before and after the fourth strand (Figure 5I–II) [14]. Mutation analysis revealed a set of CDPS-specific residues that recognized aminoacyl moieties and catalyzed their transient attachment to conserved serine residues [14,51]. Mass spectrometry analysis, X-ray crystallographic studies, site-directed mutagenesis, and affinity labeling studies have shown that the reaction intermediates were aminoacylase intermediates rather than dipeptidyl tRNA intermediates [14,46,53].

As a buffer substance in the crystallization reagent, 2-cyclohexylaminoethanesulfonic acid (CHES) and 3-(cyclohexylamino)-2-hydroxy-1-propanesulfonic acid (CAPSO) are combined in YvmC-Blic pockets in different ways (Figure 5III,IV) [14]. By analyzing how the two buffer molecules bound to the pocket, Luc Bonnefond et al. concluded that the two buffer molecules mimicked the aminoacyl moiety of the substrate and was bound to the enzyme pocket [14]: every two CHES molecules were bound to one YvmC-Blic monomer. Among them, the interaction between the first CHES molecule and the YvmC-Blic monomer was as follows: (i) the cyclohexane ring of the CHES molecule interacted with the main chains of Gly35 and Leu65 of YvmC-Blic to form a van der Waals force, (ii) the main chain nitrogen atom of the CHES molecule interacted with the side chains of Tyr180 and Glu184 of YvmC-Blic to participate in the formation of hydrogen bonds, (iii) the sulfate oxygen atom in the CHES molecule interacts with the amide group of Arg206 and Asn40 of YvmC-Blic and the hydroxyl group of Tyr204 to form a hydrogen bond. 

For the second CHES molecule: (i) the cyclohexane ring of the second CHES molecule interacted hydrophobically with the side chains of Arg206 and Pro207 of YvmC-Blic and with Met131, Glul34, and Ala135 from the symmetric molecule, and (ii) the oxygen atom of the sulfate group of the second CHES molecule formed a hydrogen bond with the main chains of Lys209 and Leu210 of YvmC-Blic [14]. In the structure where CAPSO was combined with YvmC-Blic: (i) the CAPSO cyclohexane ring had a hydrophobic interaction with the main chain of Gly35 and the side chains of Leu65 and Phe188, (ii) the nitrogen and oxygen atoms on the CAPSO main chain interacted with the carboxyl group of Glu184 to form hydrogen bonds, and (iii) an oxygen atom of the sulfate group of CAPSO formed a hydrogen bond with the amino group of Arg206 [14]. By comparing with the crystal structure of Rv2275, Luc Bonnefond et al. found that functionally conserved Tyr180 and Glu184 in YvmC-Blic were used to recognize the amide group of the Leu moiety in Leu-tRNA^Leu^, and Leu202 of YvmC-Blic was involved in substrate-specific recognition [14]. Although the combination of CHES and CAPSO with YvmC-Blic provided a basis for analyzing the binding sites of YvmC-Blic and the substrate, the composite crystallization experiment of Leu-tRNA^Leu^ and YvmC is still the gold standard to determine the substrate-specific binding site of YvmC and its interaction mode with the substrate.

Based on comprehensive experimental results, Luc Bonnefond et al. [14] proposed the mechanism of cLL synthesis catalyzed by YvmC-Blic (Figure 6). In the YvmC-Blic catalytic center, the amino group of the leucine residue from the substrate bonded to the side chains of Glu184 and Tyr180 by hydrogen bonding and, at the same time, the amide group of Asn40 polarized the ester carbonyl of leucine to promote its affinity with Ser37 hydroxyl. Subsequently, the leucine residue of the second Leu-RNA^Leu^ interacted with the carboxyl group of Glu184 through its amino group; then, the amino group of the first leucine residue attacked the carbonyl ester of the second Leu-tRNA^Leu^, which generated two leucines bound to the enzyme. The two bound leucine residues adopted the cis conformation. Finally, the amide group of the second leucine residue attacked the ester carbonyl of the first leucine residue and cyclized to cLL spontaneously [14]. However, the analysis of experimental results showed that YvmC-Blic exhibited lower substrate specificity because about 40% of the catalytic products were cyclo(*L*-Leu-*L*-Phe) and cyclo(*L*-Leu-*L*-Met) rather than the target product cLL [14].

In recent years, the number of research reports on YvmC has been very limited, especially with respect to enzymatic characterization, structural analysis and catalytic mechanism of *Bacillus* YvmC. YvmC is a key enzyme for the synthesis of cLL, which is a precursor of pulcherriminic acid, so in-depth research on YvmC will help to increase the yield of pulcherriminic acid and to promote its industrial production.

## 3. Cytochrome P450-oxidizing cLL to form pulcherriminic acid

### 3.1. Cytochrome P450

P450 was first found in mammalian tissue cells, and it was found subsequently in other eukaryotes and prokaryotes. The type of P450 found in eukaryotes was more than that in prokaryotes. Through long-term research on the prokaryote P450, key information about the structure of P450 and the mechanism by which P450 activated dioxygen and oxidized its substrate were gradually revealed [56,57,58].

Due to the many members in the P450 enzyme family, the classification of P450 is divided by the similarity of its amino acid sequences. If the similarity between two members is >40% and <55%, it is judged that they have lower identity, and so they are divided into different families (such as CYP1, CYP2, etc.). If the homology between two members of some P450 families is >55%, those families involved in sequence alignment are defined as subfamilies in a certain P450 family, distinguished by capital letters (CYP1A, etc.); individual members of a subfamily then receive consecutive numbers (CYP1A1, 1A2, etc.) [59,60]. Each member of this subfamily is then divided into bacteria, archaea, yeast/fungi, etc. based on consecutive numbers [59,60]. There are many members of the P450 superfamily, and its amino acid sequences between different species have low conservation. Interestingly, the P450 superfamily contains almost the same structural domains and very similar catalytic functions [61,62]. It is precisely because of the ability of P450 superfamily to catalyze various reactions, resulting in the production of a large number of various secondary metabolites. Moreover, the pulcherrimic acid is one of the secondary metabolites.

### 3.2. Structure and Catalytic Mechanism of Cytochrome P450 Oxidase-CypX (CYP134A1)

In 2006, Tang et al. [12] identified for the first time the pulcherriminic acid synthetic gene cluster *yvmC-cypX* in *B. subtilis* and *B. licheniformis* by using transcription analysis and bioinformatics. CypX (CYP134A1) is a member of the cytochrome P450 (P450s) blood protein superfamily. It is controlled by the *cypX* gene, which is a cytochrome P450 synthetic gene located downstream of the gene *yvmC*.

Because the crystallization of natural *B. subtilis* CYP134A1 was unsuccessful, Cryle et al. [3] obtained crystals of a selenomethionine-labeled CYP134A1 A356T mutant and used a single abnormal dispersion measurement to determine the *B. subtilis* structure of CYP134A1 A356T [3]. The structure of CYP134A1 A356T adopted the standard P450 fold, and the main R-helix structure surrounded the non-covalently attached ferro porphyrin IX (heme) part [3] (Figure 7I. The hydrogen bond on the side chain hydroxyl of the mutated Thr356 residue interacted with the main chain carbonyl oxygen of Cys353 and the side chain nitrogen of Asn352. This may be the reason why the A356T mutant formed crystals [3].

CypX structural analysis showed that the sites that usually play an important role in oxygen activation were mainly hydrophobic residues, and the usually conserved alcohol residue was replaced by an unusual proline residue [3]. In addition, an important structural element of CYP134A1 A356T is the N-terminal and central region of the I-helix (residues 229–237). Compared with other P450 structures that are close to CYP134A1 A356T, the above area traverses and occludes more heme surfaces (Figure 7). The other side corresponding to the I-helix is formed by a portion of the β-1 sheet and residues immediately adjacent to the sheet (residues 278–285 (Figure 7). The C-terminus of the F helix forms most of the ceiling above the active site (residues 158-166 (Figure 7). The Tyr391 and Thr392 residues present in the long C-terminal loop of CYP134A1 A356T help to form the upper limit of the active site. 

In different structures, Tyr391 existed in different conformations and exhibited side chain orientation, which either pointed to the active site without substrate of CYP134A1, or pointed to the middle position of the substrate’s free orientation (i.e., binding to the benzimidazole ligand) [3]. Due to the hydrogen bond between the phenol group and the glycerol oxygen atom in the crystallization reagent, the position of the tyrosine side chain depended on the existence of glycerol molecules in the active site (Figure 7) [3]. The part of the active site of CYP134A1 A356T that was most exposed to the solvent corresponded to the B-B2 loop region, which included two configurations, “open” and “closed”. In the “open” configuration, the B-B2 loop was substantially parallel to the G helix and did not affect the active site; in the “closed” conformation, the B2 helix was perpendicular to the heme [3] (Figure 7VI). Residues Leu64 and Arg67 projected into the active site cavity (residues 61–73/75 were visible in “open” and “closed” forms, respectively) [3]. A “closed” form was also observed in the structure bound to phenyl imidazole, and the B-B2 ring may be rearranged after substrate binding [3].

The cLL conversion of CYP134A1 A356T in *B. subtilis* was mediated by electron transfer, and its oxidation product appeared to correspond to the mono-oxidation product of cLL rather than to the expected final product of pulcherriminic acid [3]. The oxidation results using cLL analogs showed that there were different oxidation pathways for CYP134A1 diketopiperazine oxidation. Therefore, the exact mechanism of CYP134A1 in vivo oxidation needs to be studied further. Future studies will explore the wild-type crystal structure and oxidation products of CYP134A1 and combine the enzymatic characterization of CYP134A1 to further clarify the mechanism of the oxidation process [3]. 

Combining the identification of the biochemical characteristics of CYP134A1, Cryle et al. [3] speculated that the oxidation process of cLL was divided into three steps: first, the nitrogen atoms of the two peptide bonds of cLL was oxidized to form a nitrogen-oxygen structure; immediately afterwards, the above-mentioned nitrogen-oxygen structure removed two molecules of water; then, the six-membered ring was aromatized by intramolecular electron transfer to form pulcherriminic acid (Figure 8) [3].

## 4. Promotion of Pulcherriminic Acid Production by Metabolic Engineering 

Recent research has revealed that the synthesis of pulcherriminic acid was controlled by multiple synthetic gene clusters (Figure 9) [19,63,64]. In 2016, Randazzo et al. [17] demonstrated for the first time that a MarR-like transcriptional regulatory factor encoded by the *yvmC-cypX* adjacent gene *yvmB* bound to the *yvmC* promoter region and inhibit transcription of the *yvmC-cypX* cluster, which had a direct negative regulatory effect on the synthesis of pulcherriminic acid [17]. AbrB was an important multifunctional negative regulator in *Bacillus*, and its expression can inhibit the growth and metabolism of cells. The expression level of AbrB was growth-phase-dependent, which remained high during the exponential phase, but decreased when the cells entered the stationary phase [65,66,67]. 

By knocking out and over-expressing AbrB, Wang et al. [19] found that the synthesis of pulcherriminic acid in *B. licheniformis* was strictly regulated by AbrB-YvnA-YvmB. AbrB-deficient strains greatly increased the synthesis efficiency of pulcherriminic acid. In addition, AbrB, which is a negative regulator, inhibit the transcription of *yvmC-cypX*, and it also directly inhibit the transcription of the *yvnA* gene, which reduced the inhibitory effect of the YvnA protein on the *yvmC-cypX* operon. With multiple functions, YvnA was at the core of the regulation mechanism for synthesizing *Bacillus* pulcherriminic acid. Deletion and overexpression of the *yvnA* gene reduced the efficiency of synthesis of *Bacillus* pulcherriminic acid. Due to YvnA’s inhibitory effect on *yvmB* transcription, the *yvmC-cypX* operon transcribed only at a moderate YvnA concentration. When YvnA concentration was too low, YvmB expression increased and eventually inhibit the synthesis of pulcherriminic acid. Conversely, when the concentration of YvnA was too high, the *yvmB* and *yvmC-cypX* operons were suppressed simultaneously. This special dual-regulation system model is called “see-saw regulation” [18,63,64]. In addition, the transcription level of *yvnA* is affected by the concentration of Fe^3+^; when the concentration of Fe^3+^ in the environment was low, the transcription level of *yvnA* remained low, which resulted in a low concentration of YvnA. When the concentration of Fe^3+^ in the environment increased, the expression level of *yvnA* gene increased, and the YvnA concentration also increased [18,64]. Due to the regulatory mechanism of YvnA, there was no excessive production of protic acid to affect the cell’s own growth [68]. 

YvmA is a transporter protein that belongs to the MFS protein. Deletion of the *yvmA* gene led to a significant decrease in the production of pulcherriminic acid and an increase in the intracellular accumulation of pulcherriminic acid. The *yvmA* gene knockout cells still had a small amount of pulcherriminic acid secreted outside the cells, which indicated that the cells also had pulcherriminic acid transporters other than YvmA. The transcription of the *yvmA* gene itself was also directly negatively regulated by YvmB, and indirectly affected by AbrB and YvnA [18,19,64].

In the process of synthesizing pulcherriminic acid by *Bacillus*, the synthesis of its intermediate products is crucial to the final product. In the entire *Bacillus* system, the efficiency of the entire synthetic pathway is regulated by many genes. Regulating the production of target products through transcriptional metabolic engineering is currently the most effective strategy [69,70,71]. Precursor supply plays a key role in metabolite biosynthesis [72]. Due to the inhibition of CodY and TnrA on the ilvBHC-leuABCD operon and the negative feedback regulation of Leu on 2-isopropylmalate synthase LeuA, Leu supplementation in *B. licheniformis* may be insufficient, which indicated that insufficient substrate supply may be one of the factors that limited the synthesis of pulcherriminic acid [73,74]. In addition, the conversion of Leu to Leu-tRNA may affect the efficiency of peptide and protein synthesis [16].

Vogt et al. [75] obtained a high expression strain of L-Leu by eliminating feedback resistance and deleting the repressor LtbR for L-Leu synthesis. Zhu et al. [76] increased the accumulation of BCAAs in cells by overexpressing BrnQ, which transported branched chain amino acids (BCAA) into cells. Zhu et al. [74] enhanced the biosynthesis of Leu and other substances by deleting the *codY* gene, and they further explained the negative regulation of amino acid synthesis by CodY. Wang et al. [19] significantly increased the concentration of Leu in cells by over-expressing *ilvBHC-leuABCD* and deleting *bkdAB*, and this also increased the production of pulcherriminic acid indirectly. The *bkdAB* gene encoded the E1 subunit of a branched chain α-keto acid dehydrogenase complex, which converted branched chain amino acids (Leu, Ile, Val) into branched chain α-ketoacyl-CoA initiators (BCCS) and provided raw materials for the branched-chain fatty acid biosynthesis [19,77]. At the same time, the over-expressed aminoacyl-tRNA synthetase was also an effective strategy to promote the synthesis of cLL, which is a precursor of pulcherriminic acid. Xia et al. [78] increased the concentration of glycyl-tRNA by enhancing the expression of glycyl-tRNA synthetase; the overexpression of aminoacyl tRNA synthetase LeuS increased the formation of leucine tRNA. However, high intracellular concentrations of Leu may have triggered feedback inhibition of 2-isopropylmalate synthase LeuA [75]. In 2020, Wang et al. [19] increased the production of pulcherriminic acid to 556.1 mg/L by reconnecting the metabolic pathway of *B. licheniformis*.

## 5. Conclusions

As a class of potentially cyclic dipeptide derivatives with antibacterial effect, pulcherriminic acid has great application potential for the food, agriculture and medical industries. At present, the structures of *B. licheniformis* YvmC and *B. subtilis* CYP134A1 have been solved and, in addition, the regulatory mechanism for synthesis of pulcherriminic acid in *Bacillus* has been explored. The determination of the structure and regulatory mechanism of key enzymes for the synthesis of pulcherriminic acid by *Bacillus* laid a foundation for the industrial production and application of pulcherriminic acid.

However, the key enzymes, synthesis, and regulatory pathways for synthesizing pulcherriminic acid or pulcherrimin from *Bacillus* are still not completely clear. In view of the above problems, the future research directions for pulcherriminic acid or pulcherrimin was proposed as the following list contents:With the development of synthetic biology and metabolic engineering, biological elements from different sources are introduced into target strains to reconstruct the synthesis pathway of pulcherriminic acid, realize the heterologous expression of YvmC and CypX, and regulate the biosynthesis pathway. When the alien pathway breaks the original metabolic network of the target strain, how to maintain the metabolic balance in the cell and how to regulate the new metabolic network to achieve the best state.In order to achieve the heterogeneous and high yield of pulcherriminic acid, and meet the demands of industrialization, how to screen high activity YvmC and CypX enzyme is critical to the future research. (The catalytic mechanism of *Bacillus* cyclodipeptide synthetase YvmC and CypX is still unclear, so far).When the YvmA gene is knocked out, the pulcherriminic acid can also be transported out of the cell. Little knowledge is known about the underlying transmembrane mechanism has not been elucidated for pulcherriminic acid. How does pulcherriminic acid get outside of the cell very quickly? It is an urgent problem. What’s the balance mechanism of pulcherriminic acid and the pulcherrimin inside and outside the cell?The directed evolution technique is applied to the enzymes in the synthesis pathway of pulcherriminic acid for the screening of the key synthetases with high catalytic activity.

These are important reasons for the current limitations on the production of pulcherriminic acid. In the future, research on the synthesis pathway of pulcherriminic acid could be more focusing on the clarifying of the catalytic mechanism, structure, and function of each key enzyme, and containing the mechanism of the synthesis, secretion and regulation of pulcherriminic acid. We have reason to believe that the solutions for these problems will promote an increase in the production of pulcherriminic acid and lay the foundation for the industrial production of pulcherriminic acid.

## Figures and Tables

**Figure 1 molecules-25-05611-f001:**
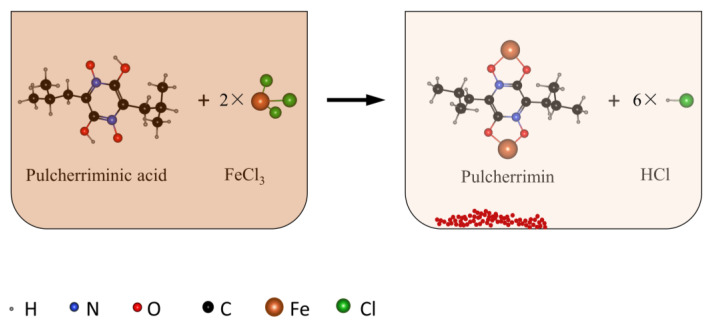
Binding process of pulcherriminic acid with iron ion (red–brown precipitate: pulcherrimin).

**Figure 2 molecules-25-05611-f002:**
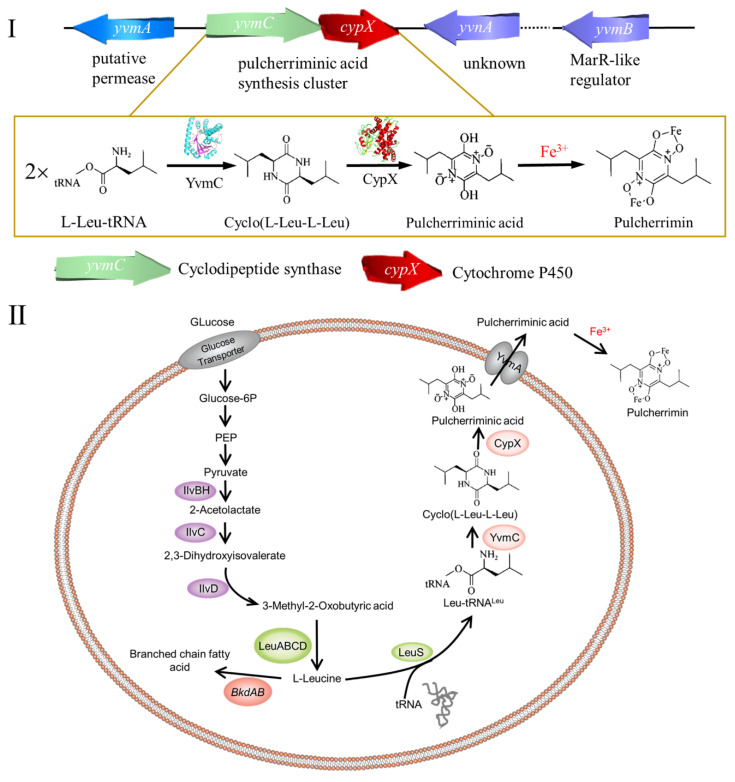
The gene cluster (**I**) and synthetic pathways (**II**) of pulcherriminic acid and pulcherrimin in *B. subtilis* [19].

**Figure 3 molecules-25-05611-f003:**
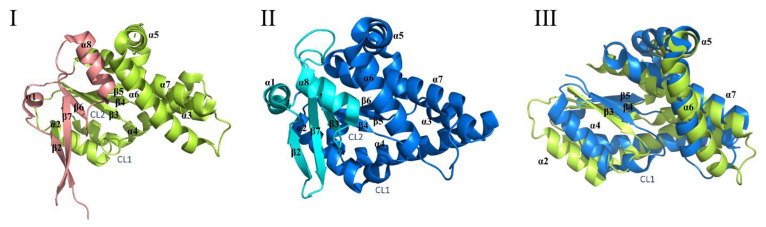
(**I**) Structure of XYP Rgry-cyclodipeptide synthases (CDPS). The first half of the Rossmann fold is shown in cyan, and the second half of the Rossmann fold is shown in limon. The protein C-terminal positions of residues 236 to 259 are not shown [51]. (**II**) Structure of NYH CDPS from *Streptomyces*. The first half of the Rossmann fold is shown in aquamarine, and the second half of the Rossmann fold is shown in marine [51]. (**III**) To better show the differences between the two subfamilies, the second part of the Rossmann fold of AlbC is superimposed to Rgry-CDPS [51], Rgry is represented by limon, and AlbC is represented by marine.

**Figure 4 molecules-25-05611-f004:**
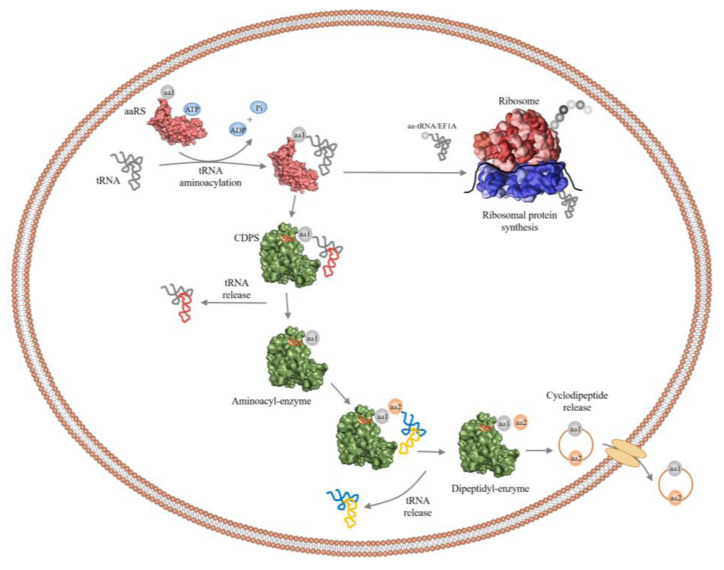
The overall catalytic mechanism of AlbC to produce cyclic dipeptides [21,38]. AlbC (green) uses two aa-tRNAs produced by aaRSs (light red) and is mainly involved in protein synthesis in the ribosome (red with blue). AlbC proceeds through a sequential mechanism that involves the formation of aminoacyl and dipeptidyl enzymes.

**Figure 5 molecules-25-05611-f005:**
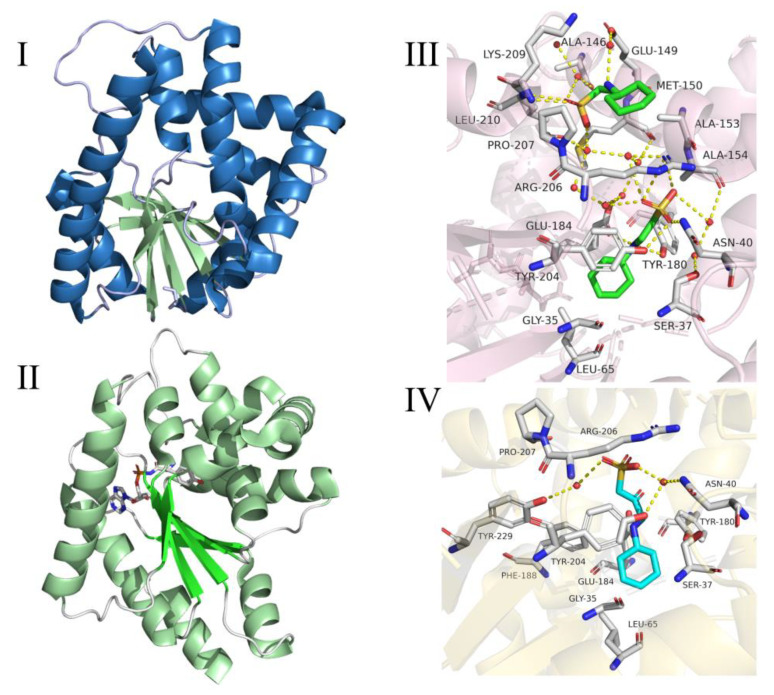
(**I**) The overall structure of YvmC-Blic. Each type of secondary structure has a different color: the ring is light blue, the helix is sky blue, and the strand is pale green [14]; (**II**) the overall structure of the yeast TyrRS catalytic domain (PDB ID 2DLC), which is similar to the overall structure (A) of YvmC-Blic. Catalytic site-bound tyrosyl-adenylate analogs were drawn using a spherical model [14]; (**III**) YvmC-Blic with two 2-cyclohexylaminoethanesulfonic acid (CHES) molecules in pockets. Water molecules appear as red spheres. Hydrogen bonds are indicated by yellow dashed lines [14]; (**IV**) YvmC-Blic with two CAPSO molecules bound to the pocket [14].

**Figure 6 molecules-25-05611-f006:**
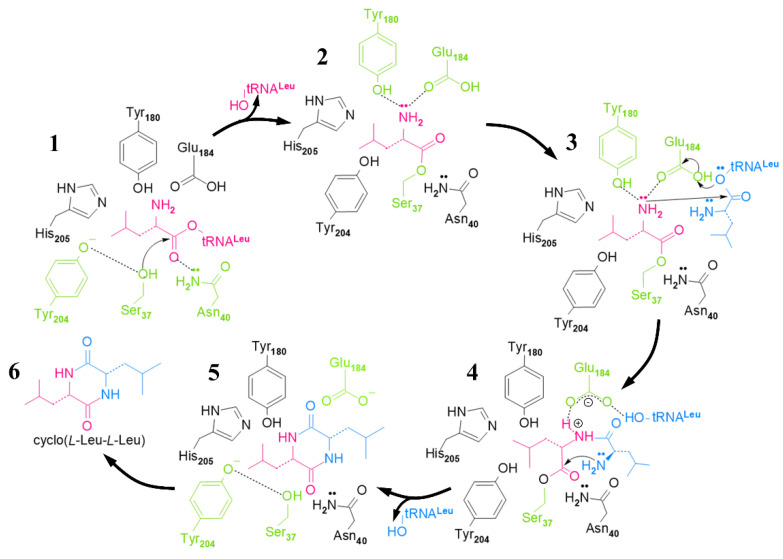
Mechanism of cyclo (*L*-Leu-*L*-Leu) (cLL) synthesis catalyzed by *B**. licheniformis* YvmC. The Leucyl moiety of the first and second combined Leu-tRNA^Leu^ is highlighted in red and blue, respectively [14].

**Figure 7 molecules-25-05611-f007:**
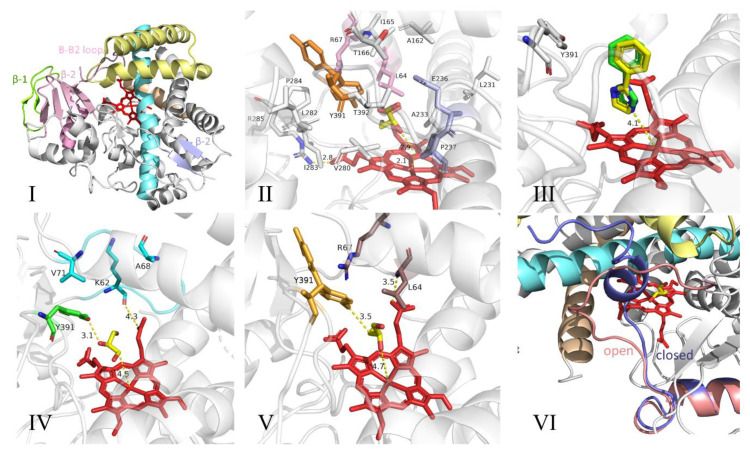
Structure and analysis of CYP134A1 A356T [3]. (**I**) The overall structure of CYP134A1 A356T (most of CYP134A1 are shown in gray, some structural motifs are colored; heme is shown in red), and the structural motifs are marked [3]. (**II**) The closed configuration of the B-B2 loop, which indicated that the active site is highly hydrophobic (B-B2 loop residues shown in pink participate in the formation of the active site pocket, and the two conformations of Tyr391 are shown in orange) [3]. (**III**) Structure of CYP134A1 A356T bound to phenyl imidazole ligand (1-phenylimidazole is shown as a green rod, 2-phenylimidazole is shown as a limon rod) [3]. (**IV**,**V**) Effects of the alternate B-B2 loop conformations on the active site pocket (open B-B2 loop and residues that form the active site shown in aquamarine, closed B-B2 loop residues that form the active site shown in darksalmon,Tyr-391 conformations shown in green for the open B-B2 loop and in bright orange for the closed B-B2 loop); for all, hydrogen-bonding distances indicated by yellow dashed lines, heme shown in red, majority of CYP134A1 shown in gray [3]. (**VI**) Alternative conformation of B-B2 ring (most CYP134A1 is shown in gray, B-B2 ring is shown in slate as closed form and salmon as open form; heme is shown in red), with selected structure Motif mark [3].

**Figure 8 molecules-25-05611-f008:**
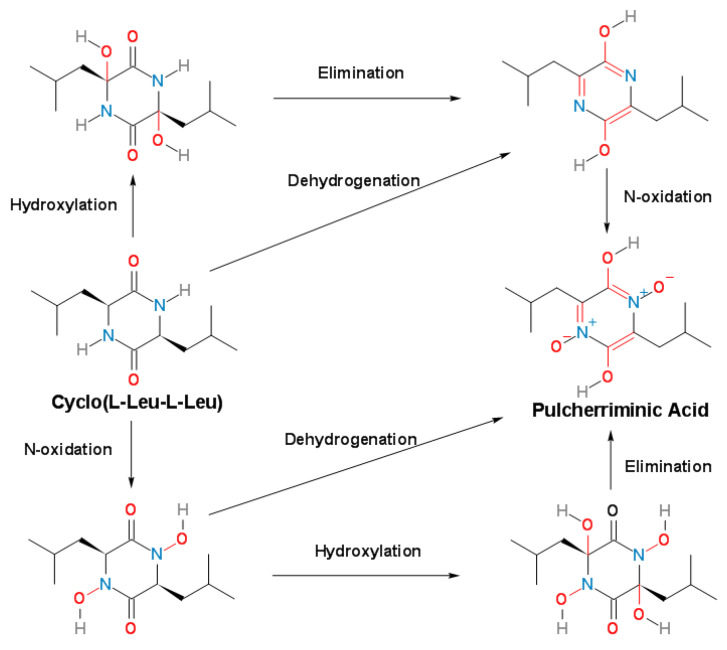
Possible mechanism of cytochrome P450 oxidase CypX that catalyzes cLL to form pulcherriminic acid [3].

**Figure 9 molecules-25-05611-f009:**
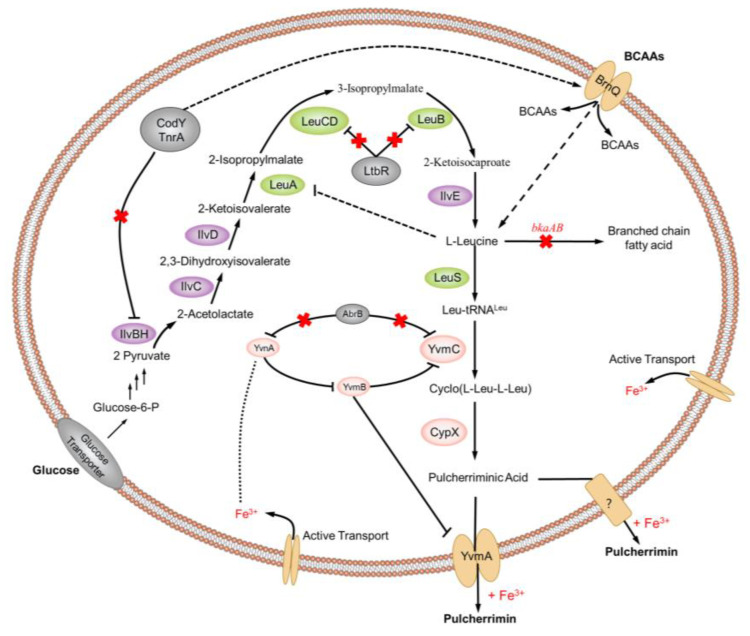
The synthesis regulation responsible for the synthesis of pulcherriminic acid in *Bacillus*. The black line with an arrow indicates the synthesis pathway of pulcherriminic acid, the black line with bars indicates inhibition, the black dashed line with bars indicates negative feedback, the red cross indicates deletion inhibition, and the red font indicates missing genes (except Fe^3+^). The dotted line with arrow indicates the promotion effect, and the dotted line indicates the effect of iron ions on protein. BCAAs, branched chain amino acids.

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
