# Peer review of "Research Progress of the Biosynthesis of Natural Bio-Antibacterial Agent Pulcherriminic Acid in Bacillus"

_molecules, 2020, doi:10.3390/molecules25235611_

Round 1

Reviewer 1 Report

Yuan, Yong et al provide a detailed review of the synthetic pathway, catalytic and regulatory mechanism of the iron chelator pulcherriminic acid and describe methods to improve its synthesis efficiency. 

The authors did a great job in collating extensive scientific data regarding pulcherriminic acid. The paper is nicely structured and easy for the reader to follow. The figures are of good quality, comprehensive and nicely complement the text. 

There are a number of minor editing changes that need to be done throughout the text to correct some typos e.g. Fig.1A MarR like regulator, line 106: catalyses, line 347: inhibit instead of inhibited etc., but overall the language is legible. Also throughout the text Bacillus should be substituted with B. e.g. B. licheniformis instead of Bacillus licheniformis. 

Apart from the above minor comments I recommend the paper for publication in Molecules. 

Author Response

  1. Fig.1A MarR like regulator

Responses: We corrected the word “reulator” to regulator in the Figure 2(I).

  1. 2. line 106: catalyses

Responses: We corrected the word “catalyzeds” to “catalyses”. Line 108.

  1. 3. line 347: inhibit instead of inhibited etc., but overall the language is legible.

Responses: All the “inhibited” are replaced by “inhibit” in the manuscript revision (Lines 345, 347, 353, 359).

4.Also throughout the text Bacillus should be substituted with B., e.g. B. licheniformis instead of Bacillus licheniformis

Responses: We use “B.” instead of “Bacillus” throughout the text e.g. “B. licheniformis” instead of “Bacillus licheniformis” (Lines 41, 46, 61, 62, 191, 193,194, 252, 280, 283, 285, 326, 351, 379, 398, 408, 409).

Reviewer 2 Report

Overall, the authors have done a comprehensive summary of the literature related to the biosynthesis of pulcherriminic acid.

However, to make a significant contribution to the field, the authors should consider the following suggestions:

  1. The review is too descriptive/summary-like. It should be made more critical of the findings on the biosynthesis of pulcherriminic acid.
  2. Although the shortcomings and future focus regarding research on biosynthesis of pulcherriminic acid were briefly and generally mentioned ( e.g. in Conclusion), it would be more valuable to pinpoint explicitly several specific directions for future research.

Other suggestions:

Line 13:

"... studying the ... catalytic mechanism ... of pulcherriminic acid ..." - the statement should be revised. It seems to suggest that pulcherriminic acid is an enzyme/catalyst, which it is not.

Line 19:

The statement " We hope that this review will ..." seems redundant. It could be more valuable if replaced with a take-home message about current research progress on pulcherriminic acid synthesis based on the authors' evaluation of the literature.

Line 35:

For structural comparison of pulcherriminic acid and pulcherrimin, showing a graphical side-by-side comparison of their structures (2D/3D) would be very useful.

Line 41:

 "... NaOH solution under strong alkaline conditions. " This statement could be made more concise.

Lines 39-42:

There should be cited references to support the information written here.

Lines 48-49:

" Leucine (Leu)...was catalyzed by a series of enzymes. " - This statement should be revised.

Lines 51-52:

"Leuyl-tRNA synthetase (LeuS) catalyzed Leu into leucine-tRNA (Leu-tRNALeu) [16]." should be revised to, e.g., "Leuyl-tRNA synthetase (LeuS) catalyzes the conversion of Leu into leucine-tRNA (Leu-tRNALeu) [16]."

Line 50:

Please recheck "... IlvCD ..." - when read against Figure 1, it is somewhat confusing as there is no ilvCD, but ilvC and ilvD in Figure 1.

Line 60:

I would suggest adding some explanation for the abbreviations in Figure 1 in the figure caption so that the figure is truly standalone. Currently, it is not possible to understand the figure without going back and forth between the text and the figure.

Lines 66-67:

"Due to the obvious bacteriostatic activity, pulcherriminic acid has great potential for use in the fields...new pesticides, fruit and vegetable fungicides..." - This statement should be revised. It seems unclear how bacteriostatic agents can be used as pesticides and fungicides.

Lines 69-71:

" The synthesis of pulcherriminic acid and its intermediates and analogs ... inhibits the growth of cancer cells. " - Are there any references/studies to support the information described in this statement?

Line 96:

" As of now, most DKPs are synthesized by..." -   The statement should be revised. "As of now" refers to based on current knowledge?

Line 106:

There are some minor typo errors here and there which should be corrected, e.g., line 106 - " catalyzeds ".

Line 119:

Please recheck to ensure all abbreviations are introduced in full at their first mention, e.g., what do " NYH and XYP" stand for?

Line 145:

" M. jannaschii " should be revised to the italic form in full "Methanocaldococcus jannaschii" - considering it is a scientific (species) name and that it is used the first time.

Line 341:

Typo error and confusing heading - " Promotione of the production of pulcherriminic acid-metabolic engineering" - please revise. Maybe to something along the lines of "Promotion of pulcherriminic acid production by metabolic engineering".

Lines 417-418:

" In the future, ... will focus on ..." should be revised to " In the future, ... could/should focus on ..."

Line 419:

It is unclear what "from a micro perspective " refers to? It would be useful to briefly explain this in the main text prior to referring to it in the Conclusion.

Author Response

Reviewer #2

Comments and Suggestions for Authors

Overall, the authors have done a comprehensive summary of the literature related to the biosynthesis of pulcherriminic acid.

However, to make a significant contribution to the field, the authors should consider the following suggestions:

  1. 1.The review is too descriptive/summary-like. It should be made more critical of the findings on the biosynthesis of pulcherriminic acid.

Responses: According to the findings on the biosynthesis of pulcherriminic acid, we partly reorganized the language to describe and generalize the findings on the biosynthesis. The research on pulcherrimic acid has not been sufficient so far, just because of the medicinal importance of pulcherriminic has attracted more and more scientists' attention.

  1. Although the shortcomings and future focus regarding research on biosynthesis of pulcherriminic acid were briefly and generally mentioned ( e.g. in Conclusion), it would be more valuable to pinpoint explicitly several specific directions for future research.

Responses:We re-read through the content of the manuscript. In view of the future research direction of pulcherriminic acid, we have summerised several possibilities in the section of conclusion presenting to the readers and corresponding researcher. ( Lines 416-430)

 Other suggestions:

Line 13: "... studying the ... catalytic mechanism ... of pulcherriminic acid ..." - the statement should be revised. It seems to suggest that pulcherriminic acid is an enzyme/catalyst, which it is not.

Responses: We revised the sentence in the abstract. ï¼ˆLines 13-15)

Line 19: The statement " We hope that this review will ..." seems redundant. It could be more valuable if replaced with a take-home message about current research progress on pulcherriminic acid synthesis based on the authors' evaluation of the literature.

 Responses: We deleted the sentence that you mentioned and rewrote a new sentence to describe the possible applications of pulcherriminic acid.” Lines 19-20.

Line 35: For structural comparison of pulcherriminic acid and pulcherrimin, showing a graphical side-by-side comparison of their structures (2D/3D) would be very useful.

Responses: We added a picture (Figure 1) to compare the differences between pulcherriminic acid and pulcherrimin. Lines 42-43.

Line 41:  "... NaOH solution under strong alkaline conditions. " This statement could be made more concise.

Responses: We revised this sentence and made it more concise ,Lines 38-40.

Lines 39-42: There should be cited references to support the information written here.

Responses:We re-edited the language and supplemented the supporting literature, Lines 38-41.

Lines 48-49: " Leucine (Leu)...was catalyzed by a series of enzymes. " - This statement should be revised.

Responses: We revised the sentence. Lines 49-50.

Lines 51-52: "Leuyl-tRNA synthetase (LeuS) catalyzed Leu into leucine-tRNA (Leu-tRNALeu) [16]." should be revised to, e.g., "Leuyl-tRNA synthetase (LeuS) catalyzes the conversion of Leu into leucine-tRNA (Leu-tRNALeu) [16]."

Responses: We revised this sentence to "Leuyl-tRNA synthetase (LeuS) catalyzes the conversion of Leu into leucine-tRNA (Leu-tRNALeu) [16]." Lines 52-53.

Line 50: Please recheck "... IlvCD ..." - when read against Figure 1, it is somewhat confusing as there is no ilvCD, but ilvC and ilvD in Figure 1.

Responses: IlvCD stands for the same type of enzyme, with similar functions, and represents together. In order to be consistent with the representation in the picture, we have revised the sentence to “These enzymes included acetohydroxy acid synthase (IlvBH), dihydroxy acid dehydratase and keto acid reductase (IlvC and IlvD), 2-isopropylmalate synthase, 3-isopropylmalate dehydrogenase and dehydratase (LeuABCD, Figure 2 II) [15].” Lines 50-52.

Line 60: I would suggest adding some explanation for the abbreviations in Figure 1 in the figure caption so that the figure is truly standalone. Currently, it is not possible to understand the figure without going back and forth between the text and the figure.

Responses: We added the legend of Figure 2. Lines 63-66.

Lines 66-67: "Due to the obvious bacteriostatic activity, pulcherriminic acid has great potential for use in the fields...new pesticides, fruit and vegetable fungicides..." - This statement should be revised. It seems unclear how bacteriostatic agents can be used as pesticides and fungicides.

Responses: We revised the sentence to “Due to the results of bacteriostatic studies, it showed that pulcherriminic acid had obvious bacteriostatic activity and could have great potential for use in fermented food, insecticides, fruit and vegetable fungicides, new pesticides and other antibacterial drugs.” Lines 70-75.

Lines 69-71: " The synthesis of pulcherriminic acid and its intermediates and analogs ... inhibits the growth of cancer cells. " - Are there any references/studies to support the information described in this statement?

Responses:  We deleted the content of this sentence and added the corresponding words.

Line 96: " As of now, most DKPs are synthesized by..." -   The statement should be revised. "As of now" refers to based on current knowledge?

Responses: We revised the sentence . Lines 100-101.

Line 106: There are some minor typo errors here and there which should be corrected, e.g., line 106 - " catalyzeds ".

Responses: We revised those minor typo errors. Lines 110,111.

Line 119: Please recheck to ensure all abbreviations are introduced in full at their first mention, e.g., what do " NYH and XYP" stand for?

Responses: Here, we rechecked the original literature describing CDPSs, in which CDPSs are divided into two subfamilies, namely the "NYH and XYP". Therefore, the two abbreviations are only symbols for the CDPSs family, and do not have a clear referential meaning. Lines 123,124. (The following is the description from Jacques IB, et al.(2015): “Comparative analysis between CDPS distribution in the tree and their sequences shows a strong correlation between the distribution and the presence of one or the other pair in the sequences. CDPSs clearly divide into two subfamilies or classes that we named ‘NYH’ and ‘XYP’ according to the ‘X40, Y202, X203’ sequence. CDPS 17 has a ‘S40, Q203’ pair and is also clearly separate from the other CDPSs in the phylogenetic tree, suggesting that it belongs to a probable third subfamily that we named ‘SYQ’. Each subfamily can be associated with a functional sub-sequence signature—whose differences are likely to reflect different ways of positioning catalytic residues—and possesses further distinguishing features. Notably, the 11 previously characterized CDPSs all belong to the NYH subfamily.” Lines 120-121. (Jacques IB., Moutiez M., et al. 2015. Analysis of 51 cyclodipeptide synthases reveals the basis for substrate specificity. Nature Chemical Biology, doi: 10.1038/nCHeMBIO.1868.)

Line 145: " M. jannaschii " should be revised to the italic form in full "Methanocaldococcus jannaschii" - considering it is a scientific (species) name and that it is used the first time.

Responses: We revised the scientific (species) name, replacing “M. jannaschii with "Methanocaldococcus jannaschii". Lines 149-150.

Line 341: Typo error and confusing heading - " Promotione of the production of pulcherriminic acid-metabolic engineering" - please revise. Maybe to something along the lines of "Promotion of pulcherriminic acid production by metabolic engineering".

Responses: We revised the chapter heading following your suggestion. Line 346.

Lines 417-418: " In the future, ... will focus on ..." should be revised to " In the future, ... could/should focus on ..."

Responses: Following this suggestion, we revised the sentence .Line 436-437

Line 419: It is unclear what "from a micro perspective " refers to? It would be useful to briefly explain this in the main text prior to referring to it in the Conclusion.

Responses: We rewrote this paragraph as description in lines 436-437.

Reviewer 3 Report

Figures have basically been copied from other people's work. No effort to properly indicate this breah of copyright. For example:

Figure 1 is modified from reference 19 Figure 1, it now includes typos and spelling mistakes

Figure 4 is directly copied from Reference 15 Figure 2.

Figure 5 is directly copied from Reference 15 Figure 5.

Figure 6 is directly copied from Reference 3 Figures 2 and 3.

Figure 7 is directly copied from Reference 3 Figure 5.

I am not going to continue reviewing this manuscript given that I consider this to be a breach of copyright and professional ethics

Author Response

Reviewer #3

Comments and Suggestions for Authors

Figures have basically been copied from other people's work. No effort to properly indicate this breah of copyright. For example:

Figure 1 is modified from reference 19 Figure 1, it now includes typos and spelling mistakes

Responses: We corrected the typos and spelling mistakes. Yes, it was modified from reference 19, because the manuscript is a review, we thought It's always right to summary and cite what our predecessors have studied .(figure 2)

Figure 4 is directly copied from Reference 15 Figure 2.

Responses: The figure is a protein structure diagram, the same amino acid site at the same location was running with the same software, Even if we want to change the viewing angle, it is unnecessary. So we chose simple modified . Figure 5.

Figure 5 is directly copied from Reference 15 Figure 5.

Responses: If we look at it carefully, the chemical structural formula of Fig. 6 has not changed, but its connotation has changed.

Figure 6 is directly copied from Reference 3 Figures 2 and 3.

Responses: The figure is a protein structure diagram, the same amino acid site at the same location was running with the same software, Even if we want to change the viewing angle, it is unnecessary. So we chose simple modified .  Figure 7.

Figure 7 is directly copied from Reference 3 Figure 5.

Responses: We appreciate the construction of the figure. Now, we have absorbed the soul of the original image, and make a big modifications.Figure 8.

 I am not going to continue reviewing this manuscript given that I consider this to be a breach of copyright and professional ethics

Responses: Thank you for your comments.We will be strict in our future research works, every works of other scientist or researcher used/cited in our manuscript we have marked out the source.

Round 2

Reviewer 2 Report

I am satisfied with the revised version and I have no further comments.

Author Response

Thank you!

Reviewer 3 Report

The figures are certainly an improvement in that they're now not copied from other papers. However, the quality of the written English in general remains very poor - spelling mistakes, grammatical errors, made up words, double negatives etc. I also still object to the tone of some of the comments of the authors with regards the field... as an example their statement "Recent reports on P450 enzymes have focused mainly on drug catalysis and metabolism in mammals; future applications and research in Bacillus remain to be explored." is simply untrue - there have been two other P450s from Bacillus crystallised and biochemically characterized (see DOI:10.1039/c4mb00665h and 10.1073/pnas.0805983105) and the role of P450s in biosynthesis is huge (see this review for examples DOI: 10.1039/c7np00063d). If the manuscript can correct the English to the point where it is not distracting to the reader and they update some of their overclaims (such as the one identified above) it will be at a point where it would be appropriate to publish. But these changes really do need to be made.

Author Response

Dear reviewer,

Thank you for your comments, we have corrected all problems which we could find out. To be honest,now the manuscript is a little bit readable according to your guidence. Please see the attachment for the point by point response.

Best regards

Dr. Jun Liu
